# Biogenic manganese oxide nanoparticle formation by a multimeric multicopper oxidase Mnx

Christine A. Romano [1], Mowei Zhou [2], Yang Song[3], Vicki H. Wysocki[3], Alice C. Dohnalkova[2], Libor Kovarik[2], Ljiljana Paša-Tolić[2] & Bradley M. Tebo[1]

Bacteria that produce Mn oxides are extraordinarily skilled engineers of nanomaterials that contribute significantly to global biogeochemical cycles. Their enzyme-based reaction mechanisms may be genetically tailored for environmental remediation applications or bioenergy production. However, significant challenges exist for structural characterization of the enzymes responsible for biomineralization. The active Mn oxidase in *Bacillus* sp. PL-12, Mnx, is a complex composed of a multicopper oxidase (MCO), MnxG, and two accessory proteins, MnxE and MnxF. MnxG shares sequence similarity with other, structurally characterized MCOs. MnxE and MnxF have no similarity to any characterized proteins. The ~200 kDa complex has been recalcitrant to crystallization, so its structure is unknown. Here, we show that native mass spectrometry defines the subunit topology and copper binding of Mnx, while high-resolution electron microscopy visualizes the protein and nascent Mn oxide minerals. These data provide critical structural information for understanding Mn biomineralization by such unexplored enzymes.

[1] Institute of Environmental Health, Oregon Health & Science University, 3181 SW Sam Jackson Park Road, Portland, OR 97239, USA. [2] Environmental Molecular Sciences Laboratory, Pacific Northwest National Laboratory, 3335 Innovation Blvd, Richland, WA 99354, USA. [3] Department of Chemistry and Biochemistry, Ohio State University, 460W 12th Ave, Columbus, OH 43210, USA. Christine A. Romano and Mowei Zhou contributed equally to this work. Correspondence and requests for materials should be addressed to C.A.R. (email: romano.christine@gmail.com) or to M.Z. (email: mowei.zhou2012@gmail.com) or to B.M.T. (email: tebob@ohsu.edu)

**M**n is an important transition metal for all life. Cycling between its reduced primarily soluble form (Mn(II)) and its oxidized insoluble forms (Mn(III,IV) oxides) is coupled in myriad ways to many elemental cycles. Mn oxides are strong oxidants, scavengers of trace elements, and important reservoirs for organic carbon[1, 2]. In surficial environments, the oxidation of Mn(II) to Mn(III,IV) oxide minerals is believed to primarily occur through direct or indirect processes carried out by diverse bacteria and fungi. The biogenic oxides produced are highly reactive and, as a consequence, Mn cycling impacts the environment more significantly than would be predicted based on Mn concentrations alone.

While research has established the importance of microbial Mn oxidation on geochemical Mn cycling, questions remain about the physiological benefits organisms derive from this process. Some bacteria may employ Mn oxidation to make them more resilient to oxidative stress[3]. This hypothesis was inspired by work demonstrating that increased intracellular Mn levels make *Deinococcus radiodurans* more resilient to radiation[4], and also by numerous studies demonstrating the ability of Mn-containing proteins and small molecules to reduce intracellular superoxide levels[5–10]. Other benefits may include disposing of excess $O_2$ by transforming it into part of an insoluble oxide mineral, forming a Mn oxide crust around a microbe which could deter predation or viral attack, or as a mechanism for degrading and gaining energy from natural organic matter[2]. This latter hypothesis is appealing since some of the enzymes that oxidize Mn(II) are related to laccases, enzymes that are involved in lignin decomposition[11, 12]. Additionally, Mn oxides themselves oxidatively degrade natural humic substances producing low molecular weight organic compounds that are potential substrates for microbial growth[2].

Recently, several enzymes involved in bacterial Mn(II) oxidation have been described. The Mn oxidases that have been characterized thus far belong to two families of proteins: the animal heme peroxidases (AHPs) and the multicopper oxidases (MCOs). Generally, these enzymes appear to catalyze the oxidation of Mn(II) to Mn(IV) via a transient Mn(III) intermediate[13–15]. The Mn(II)-oxidizing AHPs have been described in several alphaproteobacteria[13, 16, 17] and in the gammaproteobacterium *P. putida* GB-1[18]. Some AHPs appear to oxidize Mn(II) directly[13, 17], while that of *Roseobacter* sp. AzwK-3b is thought to produce $O_2^-$, which oxidizes Mn(II)[16].

MCOs are of interest to biochemists and environmental scientists because of their roles in various processes, including degradation of complex carbon sources, metal homeostasis and metabolism, and a variety of electron transfer reactions[11, 12, 19–24]. MCOs that catalyze Mn oxidation have been described in several model organisms, including species of *Bacillus*[25, 26], *Pseudomonas*[27], *Leptothrix*[28], and *Pedomicrobium*[29].

Few MCOs have been characterized for which Mn(II) is considered the primary substrate. Their study has been hampered by the difficulty in purifying these enzymes. Only recently has one such protein, the Mnx complex from *Bacillus* sp. PL-12, been heterologously over-expressed in an active form[30]. *Bacillus* sp. PL-12 is one of several marine *Bacillus* species in which Mn oxidation is carried out by the spores and not the vegetative cells[31–33]. Oxidation requires the *mnx* operon[34], a conserved feature of known Mn(II)-oxidizing *Bacillus* species[35]. Only one gene of this operon, *mnxG*, encodes a MCO. Yet a minimum of three contiguous *mnx* operon genes, *mnxEFG*, are required for activity. Their expression produces an active complex (termed Mnx) composed of MnxE, MnxF, and MnxG subunits. Mnx can oxidize both Mn(II) and Mn(III)[30]. Early SDS–PAGE and size exclusion chromatography (SEC) experiments estimated the size of Mnx at ~230 kDa. Given the theoretical molecular masses of MnxE, MnxF, and MnxG, it was suggested that Mnx contains

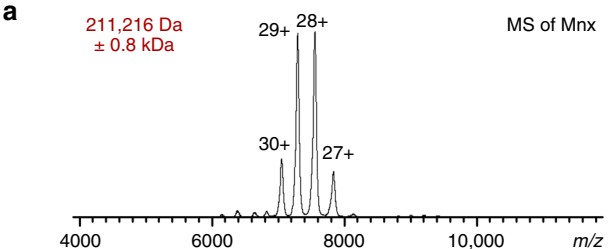

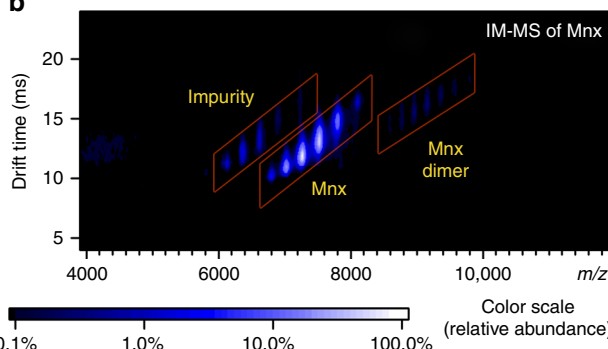

**Fig. 1** Native MS of Mnx complex. **a** Mass spectrum and **b** ion mobility–mass spectrum of the intact Mnx complex. The major species correspond to the Mnx complex with a molecular weight of 211 kDa. The impurity is a host cell protein complex that carried over during purification (Supplementary Note 2). Identical data are displayed in **a**, **b**, with **b** showing the extra dimension of separation in the vertical axis from the drift time recorded in ion mobility. The IM–MS spectrum has the *m/z* on the horizontal axis from the mass spectrometry measurement and the drift time measurement from the ion mobility separation in the vertical axis. The ions were separated based on their shape-to-charge ratio in ion mobility and arrived at the mass analyzer for *m/z* detection at discrete drift times, generating the two-dimensional spectra. The color of the spots indicates the relative abundance as shown by the scale at the bottom. The full peak width at half maximum in **a** is about 55 *m/z*, or about 1.6 kDa (±0.8 kDa) in uncertainty in mass determination

some combination of six to eight subunits of MnxE and MnxF and one copy of MnxG[30]. Expression of *mnxEF* in the absence of *mnxG* also produces a complex which binds Cu, but it is not able to oxidize Mn(II)[36]. MnxG is the only characterized MCO that requires accessory proteins, MnxE and MnxF, for successful overexpression and the only known MCO that mediates a two-electron oxidation of a metal substrate (albeit in sequential one-electron steps). It has been a research target because of chemical similarities between Mn oxides and the high oxidation states attained by the Mn-containing water-oxidizing complex of photosystem II. Studies of Mnx may inform ongoing research into the mechanisms of photosynthesis and catalytic oxygen production[37–40].

For these reasons, we investigate the stoichiometry, size, and quaternary structure of the MnxEFG complex and the structure of the nascent Mn oxide nanoparticles it produces. The similar masses of MnxE and MnxF (both ~12 kDa) prevent accurate assignment of subunit stoichiometry unless the molecular mass can be measured at high resolution. Additionally, the many metals bound to Mnx[36, 41, 42] give rise to complex spectroscopic signatures[30, 36, 42] so it is difficult to investigate individual metal cofactors or binding sites. The enzyme's fast turnover hinders efforts to capture early and intermediate stages of manganese oxide formation critical for understanding the mechanism of biomineralization[42]. Given these challenges, Mnx was an excellent candidate for alternative characterization techniques.

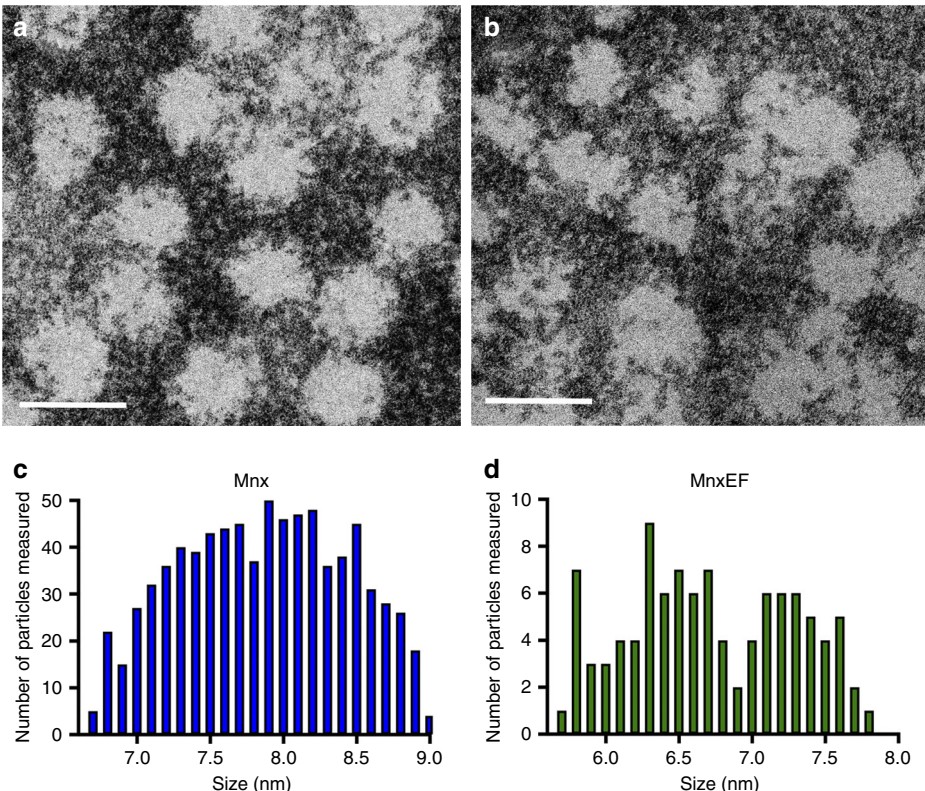

**Fig. 2** Diameters of Mnx and MnxEF particles determined by TEM. Particles were deposited onto a 300 mesh gold TEM grid with ultrathin carbon film on lacey carbon support film and counterstained with nanoW. **a** Representative image of the Mnx complex. **b** Representative image of MnxEF particles. Scale bars are 10 nm. Size distribution of **c** Mnx and **d** MnxEF particles from TEM measurements. Histograms show the number of particles counted within a specific diameter range (<6.8, between 6.8 and 6.9, and so on) for both Mnx (**c**) and MnxEF (**d**)

We use native mass spectrometry (MS) to obtain an accurate molecular mass for the intact Mnx complex and to assess its subunit and metal-binding stoichiometry. The complex exhibits unexpected heterogeneity, requiring additional scrutiny. We employ surface-induced dissociation (SID), a dissociation method that dissects the native Mnx complex into smaller subunits while maintaining bound metals. We also wondered how Mnx avoids becoming encrusted in the nanoparticles it makes. To address this question, we apply scanning transmission electron microscopy (S/TEM) to characterize solutions of Mn minerals formed by Mnx. We establish the size and multimeric structure of the protein, that MnxE, MnxF, and MnxG each differ in their affinity for Cu, and examine the formation of the incipient layered Mn oxide mineral. From these data we develop a structural model of the Mnx complex and gain insights into the mechanism of formation of nanoparticulate Mn oxide minerals.

## Results

**Defining the size of the Mnx complex by native MS and TEM.** Relative to the ~230 kDa previously estimated by SDS–PAGE and SEC, native MS of the Mnx complex (electrosprayed in 100 mM ammonium acetate, pH 7, to preserve noncovalent assembly) allowed a more accurate molecular mass determination of 211,216 Da ± 0.8 kDa. The uncertainty is due to the large compositional heterogeneity of the protein complex. Mnx dimer was observed at very low abundance (Fig. 1). It probably originated from nonspecific association in the electrospray[43]. Top-down MS of Mnx under denaturing conditions yielded an accurate molecular mass for each subunit (Supplementary Fig. 1, Supplementary Methods) and indicated a disulfide bond in MnxE and a truncation at the N-terminus of MnxF. The intact MnxG

was not detected under denaturing conditions, so the theoretical mass (Supplementary Note 1) was used in our calculations to estimate that the complex contains one MnxG and some combination of six MnxE and MnxF subunits. However, the exact number of MnxE and MnxF bound cannot be determined exactly due to heterogeneity and limited resolution (Supplementary Fig. 2). The ion mobility–MS (IM–MS) of the Mnx complex (Fig. 1b) also provided drift time measurements, which can be converted into the collisional cross section (CCS) of the ions (Supplementary Table 1).

High resolution S/TEM of the intact Mnx complex and MnxEF was performed to obtain estimates of the sizes of the complexes (Fig. 2). The average diameter of the Mnx complex and MnxEF construct were measured to be 7.9 ± 0.6 nm and 6.8 ± 0.6 nm, respectively. The size of the observed MnxEF construct suggests that MnxE and MnxF can form an assembly which associates with MnxG. Yet the number of MnxE and MnxF bound within the complex remained unresolved. IM–MS data were not shown for the MnxEF construct due to its instability and precipitation in MS-friendly buffers. Attempts to partially destabilize the Mnx complex in solution and detect released subcomplexes were unsuccessful (data not shown). We were, however, able to separate the putative MnxEF assembly in situ from the 211 kDa Mnx complex by gas-phase activation and determine its quaternary structure by IM–MS.

**Dissection of the Mnx quaternary structure by SID.** The IM–MS spectrum of the isolated 29+ Mnx complex shows almost exclusively a single species (Fig. 3a). Collision-induced dissociation (CID) (Fig. 3b) yielded MnxE and MnxF monomers at $m/z \sim 2000$, leaving behind the stripped complex in the high

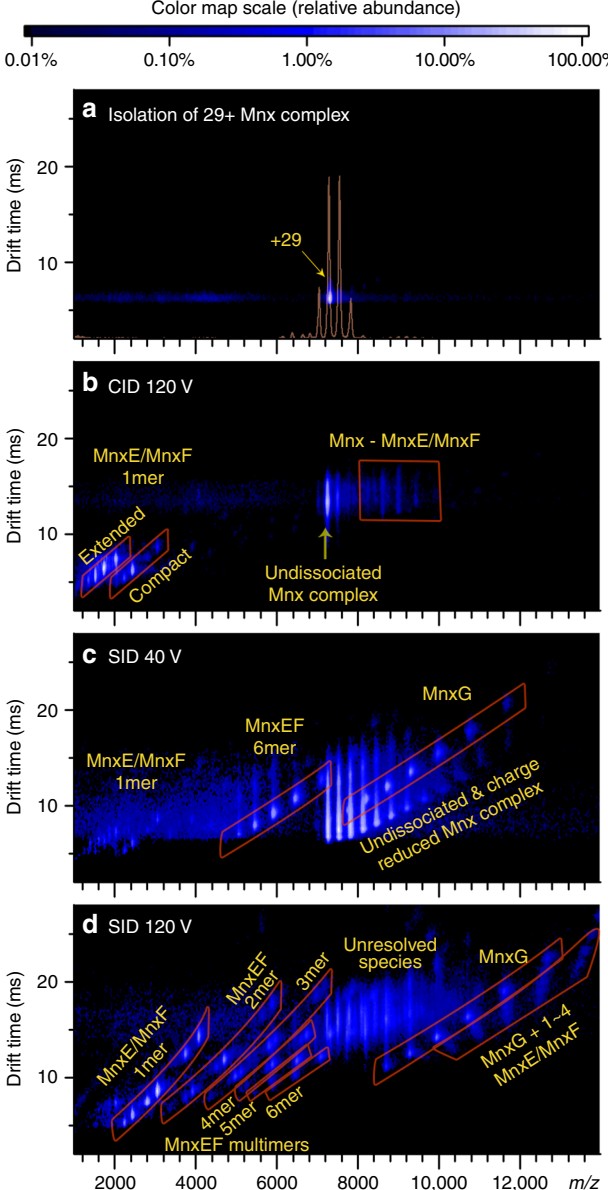

**Fig. 3** Dissecting Mnx in the gas phase for characterization of quaternary structure. Ion mobility mass spectra of **a** the isolated 29+ Mnx complex, **b** CID at 120 V, **c** SID at a low collision voltage of 40 V, and **d** SID at a high-collision voltage of 120 V. The color of the spots represents the relative abundance, with the color scale above the figure. Figure 1a was overlaid in **a** as the light orange trace to better display the single charge state of Mnx that was isolated. When the Mnx complex was activated in the collision cell pressurized with argon in CID **b**, monomers of MnxE and MnxF were released (around $m/z$ 2000), leaving behind the stripped complex (Mnx complex losing one MnxE or one MnxF, around $m/z$ 9000). Upon activation of SID at 40 V **c**, the complex dissociates into MnxEF hexamer (mostly $E_3F_3$) and MnxG (highlighted in red parallelograms), suggesting that the Mnx complex is comprised of a MnxEF hexamer attached to a MnxG monomer. At higher activation energy **d**, more extensive dissociation of the MnxEF hexamer and Mnx complex into smaller subcomplexes can be observed

$m/z$ region ($m/z \sim 9000$). The pattern of single monomer ejection is a commonly observed behavior in CID for many protein complexes[44, 45]. Previous studies have attributed this behavior to the unfolding of the exiting monomer induced by slow-heating with incremental low-energy collisions of the large protein complex with neutral collision gas[45, 46]. The ejected subunits

helped confirm the composition of the complex, but yielded only limited information about the quaternary structure. No additional subunits were released at significant abundance for the whole range of accessible collision energies (Supplementary Fig. 3).

In contrast, SID, in which the protein complexes are collided with a surface target, preferentially breaks the weakest inter-subunit interface because of the rapid and high-energy activation step, generating subcomplexes that reflect the native protein structure and subunit connectivity[44, 47]. Low-energy SID (Fig. 3c) resulted in dissociation of the Mnx complex into a 70.7 kDa species (at $m/z \sim 6000$) and a 139 kDa species (at $8000 < m/z < 12,000$) which were identified as MnxEF hexamer and MnxG based on agreement with the theoretical masses. The remaining peaks (not highlighted in Fig. 3c) at $7000 < m/z < 12,000$ corresponded to the undissociated Mnx complex. We were able to confidently assign the hexamer to be $MnxE_3F_3$ after removing the majority of bound copper in MnxE/MnxF with ethylenediaminetetraacetic acid (EDTA) (Supplementary Table 2, Supplementary Fig. 4, and Supplementary Note 3). While we observed weak signal from MnxE and MnxF monomers in the low $m/z$ region (at $m/z \sim 2000$), the major SID products corresponded to $MnxE_3F_3$ hexamer and MnxG, implying that the Mnx complex is comprised of one MnxG bound to a hexamer of three MnxE and three MnxF.

Higher-energy SID yielded more extensive dissociation and appearance of other types of subcomplexes (Fig. 3d). Below $m/z = 7000$, all types of MnxEF multimers, from hexamers to dimers, were resolved into distinct regions in addition to the MnxE and MnxF monomers as highlighted by the red-outlined parallelograms in the spectrum. This complete series of monomers to $n$-mers is reminiscent of previous SID data of cyclic protein complexes[44], suggesting $MnxE_3F_3$ may also be cyclic. The outlined regions above $m/z = 8000$ corresponded to MnxG and the Mnx complex losing several copies of MnxE and/or MnxF (in other words, MnxG attached to a few MnxE/MnxF monomers). The features in the middle of the spectrum ($m/z \sim 8000$ and drift time $\sim 15$ ms) are mainly the undissociated precursor overlapped with some of the released MnxEF multimers and MnxG. Nonetheless, the masses of the MnxEF multimers below $m/z = 6000$ enabled accurate assignment of MnxE:MnxF ratios for the released subcomplexes (Supplementary Table 2, Supplementary Fig. 4). SID products indicated that the major multimers in the complex were $E_1F_1$, $E_1F_2$, $E_2F_1$, $E_2F_2$, $E_2F_3$, and $E_3F_3$, each carrying a number of Cu atoms (unless treated with EDTA). Most pieces observed have a ratio of MnxE:MnxF near 1:1, i.e., no $E_2$, $E_3$, $F_3$, $E_3F_1$ were observed, and $F_2$ and $E_1F_3$ were only detected at trace levels and possibly originated from secondary dissociation of larger multimers. Therefore, it is reasonable to propose that the $E_1F_1$ dimer is a basic unit of a $E_3F_3$ hexamer. Given the symmetry of protein assembly as reported in the literature[48], and the released subcomplexes detected experimentally, the most reasonable quaternary structure of the $MnxE_3F_3$ hexamer consists of alternating MnxE and MnxF monomers with threefold symmetry (Fig. 4a). The hexamer then binds MnxG to form the Mnx complex. Mapping the interactions between MnxE and MnxF using chemical crosslinking proved to be challenging, presumably due to low crosslinking efficiency. Still, the subunit connectivity determined from SID data, combined with protein modeling and docking simulations[49–51] led to a tentative structural model of the Mnx complex, shown in Fig. 4b (more details in Supplementary Fig. 5).

**Metal-binding stoichiometry of the MnxE/MnxF subunits.** While unfolding in CID often results in loss of metals bound to proteins[45, 52, 53], SID often causes minimal unfolding, so these metals are more likely to remain bound[44, 45, 52, 53]. Consequently, SID provided a unique means of determining the metal-binding

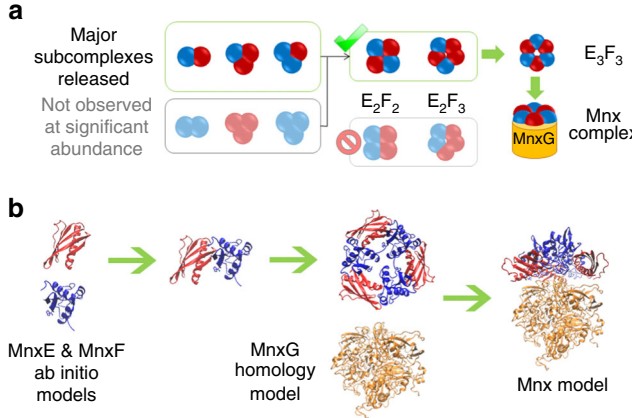

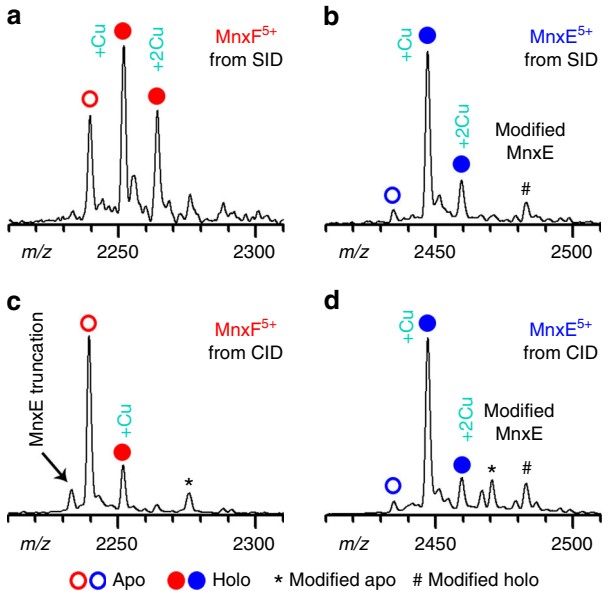

**Fig. 4** Building a structural model of Mnx based on the subunit connectivity determined from native MS experiment. **a** Determination of the topology of Mnx complex based on observed subcomplexes released in SID. The major species observed for the released MnxEF dimers are $E_1F_1$, and for the released trimers are $E_1F_2$ (~2/3 of the trimer in the spectrum), and $E_2F_1$ (~1/3 of the trimer in the spectrum). The lack of a strong signal for homo-dimers and homo-trimers ruled out some possible configurations for the observed $E_2F_2$ and $E_2F_3$ subcomplexes, leaving a symmetric hexamer structure being the most reasonable solution. **b** With the topology proposed based on SID data, computational models of the subunits can be docked to generate possible models of Mnx complex with additional constraints from CCS and surface labeling experiments (Supplementary Methods and Supplementary Fig. 5)

**Fig. 5** Examining metal-binding stoichiometry of in situ dissected MnxE and MnxF subunits. Zoomed-in view of the mass spectra for monomers released from Mnx, MnxF and MnxE, by SID (**a**, **b**) or CID (**c**, **d**). Peaks corresponding to the apo protein are labeled with open circles; copper bound proteins are labeled with filled circles. Asterisks (*) annotate protein peaks thought to arise from artificial modification during protein expression and purification; the pound sign (#) represents the modified protein bound to copper. MnxE primarily binds one copper with strong affinity. MnxF binds 0–2 copper; the stoichiometry is variable depending on sample conditions, implying a weaker affinity to copper than MnxE. In CID, most of the weakly bound copper on MnxF was lost after the subunits were released from the complex due to protein unfolding. Because minimal unfolding of the subunits occurs in SID, the weakly bound ligands can be preserved (see Supplementary Fig. 6 and Supplementary Note 4 for more details)

stoichiometry within individual subunits of the Mnx complex. A zoomed view of the SID and CID spectra of the released MnxE and MnxF+5 charge state ions indicates that the MnxF monomer released in SID retained the bound copper ions (Fig. 5a), while the MnxF monomer released in CID had lost most of its bound metal ions (Fig. 5c). Different behavior was observed for the released MnxE monomer, where the majority of the protein bound one copper under both SID and CID conditions (Fig. 5b, d). The spectra show additional, low abundance peaks larger in mass than the Cu-bound species. These peaks were investigated by a separate SID experiment on a modified high-resolution Fourier Transform Ion Cyclotron Resonance mass spectrometer. They are MnxE/F monomers modified by gluconoylation, 176 Da, a known artifact for proteins expressed in *E. coli*, and/or 4-(2-Aminoethyl)benzenesulfonyl fluoride hydrochloride (183 Da), a protease inhibitor used in the early stages of purification[54]. Similar peaks were observed in the CID and SID spectra for other charge states of released MnxE and MnxF monomers (Supplementary Fig. 6). Overall, MS data suggest MnxE binds copper with a high affinity and a primarily 1:1 stoichiometry. MnxF can bind multiple copper atoms but exhibits lower affinity to copper than MnxE. Differences in metal-binding behavior between MnxE and MnxF may be due to the location of the Cu-binding sites within the protein (Supplementary Note 4). Based on the mass of the MnxEF multimers released in SID, we concluded that in the MnxEF hexamer, each of the MnxE and MnxF monomers bind more than one copper on average (Supplementary Table 2). The heterogeneity of the protein (e.g., variable protein modifications and metal-binding stoichiometry) and the limited resolution did not allow the metal binding on the released MnxG to be accurately determined. MnxG showed a broad peak in *m/z* and an experimental mass about 1 kDa larger than expected from its sequence. The total copper load on Mnx was estimated to be 10–15 atoms per molecule by inductively coupled plasma optical emission spectrometry[41]. With $MnxE_3F_3$ already binding nearly ten

copper, the 1 kDa mass shift cannot be satisfactorily explained only by bound copper, suggesting the existence of additional ligands or unknown modifications that require further study.

**Visualizing early stage Mn nanoparticles generated by Mnx.** High-resolution S/TEM was performed to observe the structure of the nascent Mn oxide particles (Fig. 6). The particles were examined after 30 min of reaction time. The mineralized Mn oxides first appear as nanoparticulate crystallites ranging from 1 nm to about 10.5 nm in length with an average size of $2.5 \pm 1.1$ nm. The microcrystalline nature of the particles is evidenced by the well-defined lattice fringes. Electron energy loss spectroscopy (EELS) analyses (Supplementary Fig. 7) confirm the presence of oxidized Mn. By comparison with an $MnCl_2$ standard we can conclude that the oxidation state corresponds to $3^+$ or $4^+$ (~ 1.6 eV increase). The crystallites appear to merge into larger polycrystalline nanoparticles (Fig. 6). The overall formation mechanism resembles that described previously for iron oxide nanoparticle growth[55].

**Discussion**

The combination of state-of-the-art MS and electron microscopy has allowed us to establish a structural model of the previously uncharacterized Mnx complex with a diameter on the order of 8 nm, a molecular mass of 211 kDa, and a stoichiometry of $MnxGE_3F_3$ with the MnxE and MnxF subunits forming a hexamer arranged in alternating fashion (Fig. 4). In particular, we were able to determine the stoichiometry of how MnxE and

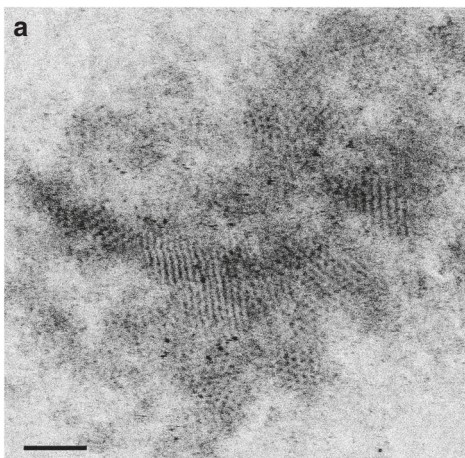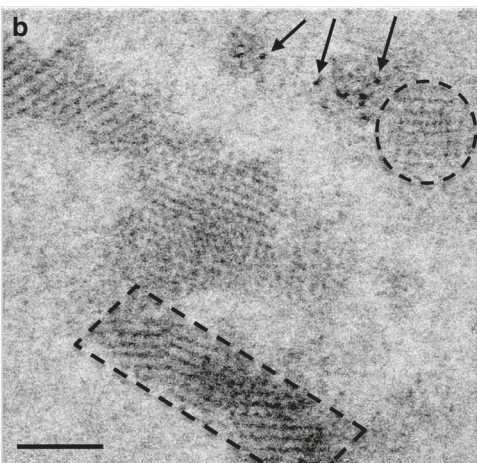

**Fig. 6** Representative TEM images of nascent Mn oxide nanoparticles produced by Mnx. The reaction proceeded for 30 min at a substrate to enzyme molar ratio of 5000:1. **a** Agglomerated forms of nanoparticles (crystallites). Individual crystallites are ~1.1–4.2 nm in length, with an average diameter of 2.3 ± 0.7 nm. **b** Atomic resolution TEM revealed three distinct forms of nanocrystalline material: amorphous clusters consisting of individual Mn atoms (arrows), individual nanocrystals (dashed circle), and agglomerations of crystallites forming lath-like crystals (dashed rectangle—notice the $MnO_x$ atomic rows). Scale bars are 2 nm

MnxF, neither of which has known homologs in the available databases, assemble with each other and with MnxG in a single-native MS experiment. Although further studies are required to solve the structure of the Mnx complex at high resolution, the techniques used to characterize Mnx represent a powerful advance, and could be applied to other metalloprotein complexes, thus helping elucidate the structure of many enzymes that are intractable by other structural methods.

Notably, prior experiments performed on MnxEF examined it as a strep-tagged construct that, because it was not attached to MnxG, may represent a non-native state[36]. The SID technique described herein produced untagged MnxE and MnxF in situ by dissociating them in the gas phase from the active Mnx complex. In SID, MnxE and MnxF are released from the complex as compact, folded structures, so the experiment provides key information about different biochemical features in regard to metal binding. The SID data show that MnxE primarily binds one copper, and MnxF is able to bind up to three copper ions albeit with lower affinity than MnxE. The differences in copper-binding stoichiometry and affinity observed for MnxE and MnxF indicate they likely perform different functions within Mnx. This work showed that MnxE contains a disulfide bond and, based on spectroscopy data, the MnxEF construct may bind heme and contain a type 2 copper center coordinated by three histidines[36]. These data suggest MnxE and MnxF may be redox active and participate in metal or electron transfer.

Aside from Mnx, no MCOs have been characterized for which minerals are thought to be the primary enzymatic product. The minerals have to be released in such a way that the Mnx complex does not become encumbered by Mn oxides[56]. In the case of one of the most famous mineral—forming enzymes, ferritin, oxidized products are shuttled to a central cavity[57, 58]. In contrast, in *Roseobacter* sp. AzwK-3b, a secreted AHP generates superoxide, which then generates Mn oxides[16] so the Mn oxides are less likely to associate with the enzyme. The S/TEM images of Mnx-generated Mn oxides show the Mn atoms arranged into crystallites with discernible lattice fringes; these crystallites then coalesce into larger particles (Fig. 6). These observations suggest that Mn oxide particles are released from the Mnx complex as clusters containing a few Mn and oxygen atoms. They then aggregate into larger minerals while in solution. Unlike ferritin[59], the Mnx complex does not appear to actively pattern or arrange

the Mn oxide minerals within the protein. This process may be similar to that described for *Roseobacter* sp. AzwK-3b species, in which Mn oxides are first observable within 20 h as 10–100 nm colloidal Mn oxides that then aggregate into larger, particulate minerals whose sizes vary with experimental conditions[60, 61]. This aggregation may happen through random interactions, or the Mnx complex may serve as a template onto which Mn particles can agglomerate.

In nature, manganese oxides act as sponges for trace elements and have the potential to react with other chemicals and trace nutrients or be microbially coupled with other elemental cycles. Nanoparticles' reactivities depend on their size, age, structure, and solution conditions[62]. Our S/TEM results suggest that biogenic Mn oxide particles in the environment begin as extremely small nanoparticles that would have very high surface areas and reactivities. While the high ionic strength of seawater would promote eventual aggregation and settling, the nanoscale size of the particles, their reactivity, and their oxidative capacity would strongly impact other biogeochemical processes coupled to the Mn cycle and, thus, would be globally important.

## Methods

**Protein expression and purification**. Protein was purified using plasmid constructs and purification methods described previously[30, 36, 42]. To isolate untagged protein, the *mnxDEFG* construct was cloned onto a pTXB1 plasmid (kindly provided by the group of N. Blackburn, Oregon Health & Science University). For strep-tagged protein, the same construct was cloned into the Strep-Tagged pASK/IBA3 plasmid (IBA Life Sciences).

For cell growth, a plasmid containing the *mnxDEFG* construct was transformed into *E. coli* BL21 (DE3) and grown at 37 °C to an $OD_{600}$ ~ 0.5 in Luria-Bertani broth containing 0.2 mM $CuSO_4$, 10 mM Tris-HCl pH 7.5 (Tris=2-Amino-2-hydroxymethyl-propane-1,3-diol), and 100 mg/L ampicillin. The temperature was then lowered to 17 °C by cooling the culture on ice or in a refrigerated shaker. Transcription of the *mnx* genes was induced by addition of anhydrotetracycline to a final concentration of 0.2 mg/L (pASK construct) or 1 mM IPTG to a final concentration of 1 mM (pTXB1 construct). The cells continued to shake and express for 16–20 h. $CuSO_4$ was added to a final concentration of 2 mM and the shaking function was stopped for at least 22 h more to allow for the microaerobic uptake of copper ions in the *E. coli* cytoplasm as described in Durao et al.[63]

The cells were then collected and suspended in buffer. For the untagged construct, the buffer was hydrophobic interaction chromatography (HIC) start buffer (20 mM Tris-HCl pH 7.5,1.25 M NaCl) supplemented with 10 mM $CaCl_2$, 1 mM CuSO4, and an EDTA-free SIGMA*FAST*Protease Inhibitor Cocktail Tablet (Sigma). For the tagged sample, streptactin equilibration buffer was used (100 mM Tris pH 8.0, 150 mM NaCl), amended with 10 mM $CaCl_2$, 1 mM $CuSO_4$, and an EDTA-free SIGMA*FAST*Protease Inhibitor Cocktail Tablet. Resuspended cells

were lysed by French Press. The cell lysate was clarified by heat denaturation at 70 °C for 15 min followed by centrifugation at $15,000 \times g$ 4 °C 30 min to separate debris. The supernatant was filtered through a 0.4 μm pore polyvinylidene fluoride (PVDF) filter.

The clarified lysate was then passed through a series of columns. The untagged sample was loaded onto a gravity flow column packed with Phenyl Sepharose 6 Fast Flow resin (high sub, GE Healthcare Life Sciences) in HIC start buffer. It was washed with HIC wash buffer (20 mM Tris-HCl, pH 7.5, 0.5 M NaCl) and then eluted using HIC elution buffer (20 mM Tris-HCl, 50 mM NaCl). The eluted sample was concentrated and loaded onto an FPLC-based HiTrap Q HP column (GE Healthcare Life Sciences) where it was purified in 20 mM Tris, pH 7.5 using a NaCl gradient of 50 mM to 1 M.

The tagged sample was mixed with Strep-Tactin Superflow Plus (Qiagen) resin and slowly rotated for 1 h at room temperature. By gravity flow, the unbound protein fraction was removed and the resin was washed with 20 CV Streptactin equilibration buffer. The Mnx protein was eluted with 5 CV equilibration buffer plus 2.5 mM D-Desthiobiotin and the column was regenerated with 15 CV equilibration buffer plus 1 mM 2-(4-hydroxyphenylazo)benzoic acid.

For both the tagged and untagged protein, the eluted protein was next concentrated to <1.5 mL on 100 kDa molecular weight cutoff filtration units (Millipore) for loading onto HiPrep 16/60 Sephacryl S-200 High Resolution gel filtration column (GE Healthcare) equilibrated with equilibration buffer (20 mM HEPES (HEPES = 2-[4-(2-hydroxyethyl)piperazine-1-yl]ethanesulfonic acid) pH 7.8, 50 mM NaCl, 5 % D-glucose (weight/volume)) at 4 °C. All buffers up to this point are supplemented with 50 μM $CuSO_4$ to avoid copper leaching by Tris. Mnx-containing fractions were collected, pooled, concentrated, and dialyzed three times for at least 2 h each in 1 L dialysis buffer (20 mM HEPES pH 7.8, 50 mM NaCl) at 4 °C. The protein was quantified by the Thermo Scientific Pierce bicinchoninic acid (BCA) protein assay. This as-isolated oxidized protein complex (denoted as Mnx) was then flash frozen in liquid nitrogen and stored at –80 °C until use.

**Electron microscopy of protein particles**. For TEM imaging, samples of protein were diluted to a concentration of 3.8 mg/mL in dialysis buffer and deposited onto a TEM grid (100 mesh copper grid coated with formvar and carbon, Electron Microscopy Sciences), and negatively stained with NanoW (Nanoprobes). They were examined in a Tecnai T-12 transmission electron microscope (FEI). The ultrastructural observations were performed with an aberration corrected FEI Titan 80–300 transmission electron microscope. The observations were made in S/TEM mode using a high angle annular dark field (HAADF) detector. The probe convergence angle was set to 18 mrad and the inner detection angle on the HAADF detector was set to 52 mrad. For high resolution TEM, samples were diluted to 70 μg/mL (Mnx complex) or 35 μg/mL (MnxEF). When protein samples were visualized in the absence of Mn, an aliquot of NanoW (Nanoprobes) was added as a negative stain. Particle sizes were measured using ImageJ software[64]. Over 100 particles of MnxEF and over 1000 particles of Mnx complex were measured.

**Electron microscopy of protein and Mn oxide particles**. Mnx complex (431 nM) was mixed with $MnCl_2$ at a final concentration of 431 μM or 2.15 mM (1000× and 5000× molar ratio substrate to enzyme, respectively). Samples of Mn oxides were produced by the Mnx complex over a time course of ~30 min Each reaction had a final volume of 110 μL. In total, 5 μL of each reaction suspension was pipetted onto a 300 mesh gold TEM grid with ultrathin carbon film on lacey carbon support film (Ted Pella, Inc.). They were then examined by high resolution S/TEM/EELS for ultrastructure, crystallinity, and oxidation state of the Mn. Particle sizes were measured using ImageJ software.

**Native mass spectrometry**. The purified Mnx protein complex was buffer exchanged into 100 mM ammonium acetate with Micro Bio-Spin P-6 gel columns (BioRad, Hercules, California, USA). The concentration of the protein after buffer exchange was estimated with a NanoDrop 2000 C spectrophotometer (Thermo Fisher scientific, Wilmington, Delaware, USA) with a predicted extinction coefficient of 214,840 per M/cm at 280 nm. The protein complex was loaded on glass capillaries pulled in house, at a volume of about 5 μL and a concentration of about 1 mg/mL. The MS data were acquired on a modified G2s ion mobility mass spectrometer (Waters Corporation, Manchester, United Kingdom). The details of the modification to incorporate the SID device can be found in previous reports[44]. Briefly, the modified mass spectrometer allows the electrosprayed native protein complexes to be activated via either surface collision (SID) or neutral gas collision (CID) for dissociation before ion mobility separation. Activation in the trap ion guide of the instrument occurs after quadrupole $m/z$ selection and before IM. The final step involves mass detection by the time-of-flight mass analyzer. The detailed instrument settings are listed in Supplementary Table 3. Mass calibration was achieved with cesium iodide cluster ions up to $m/z = 8000$ and extrapolated into the full mass range up to $m/z = 14,000$.

**Data availability**. The data that support the findings of this study are available from the corresponding authors upon reasonable request.

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

## Acknowledgements

We thank Jing Yan and Prof. Si Wu for helping with some of the MS experiments. The project was funded by the National Science Foundation CHE-1410688 to B.M.T., the National Institute of Health (NIH R01GM113658) to V.H.W., and an NSF Postdoctoral Research Fellowship in Biology Award ID: DBI-1202859 to CAR. A portion of the research was supported by the Environmental and Molecular Sciences Laboratory (EMSL), a DOE Office of Science User Facility sponsored by the Office of Biological and Environmental Research and located at Pacific Northwest National Laboratory.

## Author contributions

C.A.R. and M.Z. contributed equally to the work and wrote the manuscript with inputs from all authors. C.A.R. prepared the Mnx protein. M.Z. and Y.S. performed the MS experiments and analyzed the data. C.A.R., A.C.D., L.K., and B.M.T. performed the microscopy experiments and analyzed the data. All authors were involved in discussion and design of experiments.
