## [Peer Review File · Nature Communications]

Reviewers' comments:

Reviewer #1 (Remarks to the Author):

The manuscript by Romano et al. describes a study that examines the protein structure, stoichiometry and resulting Mn oxides from a complex Mn-oxidizing protein complex (Mnx). The manuscript was very well written and the flow was logical. The data presented was well justified and the authors' conclusions are within the scope of their data. Bacterial Mn-oxidizing proteins have been extremely difficult to study. To date, the Mnx complex is the only via option to get a high quality, purified protein. Thus, this study is very unique as it presents the best structural data on a Mn-oxidizing protein (even though many bacteria are able to oxidize Mn). In addition, the data helps provide important insight into how a spore-forming bacteria is able to generate Mn oxide particles via a protein without that protein getting engulfed with solid phase particles.

Reviewer #2 (Remarks to the Author):

The manuscript by Romano et al. addresses a complex and demanding biological system, Mnx enzyme assemblies which synthesize Manganese Oxide nanoparticles. "Standard" structural or molecular approaches are not expected to give straightforward insights into the structure and function of these complexes, and the choice of cutting-edge MS approaches combined with EM is appropriate.

I think this work demonstrates also how polydisperse and otherwise untractable complexes can be investigated, and sets a good example for how novel MS approaches (in this case SID) can be utilized to unravel complex composition and architecture.

I am convinced of the findings and agree with the data interpretation, with the (ample) evidence provided, but feel that maybe the inclusion of some minor, additional evidence from other, more readily available MS data would round the story nicely and confirm the otherwise somewhat vague and tentative conclusions, particularly regarding the alternating arrangement of MnxE/F subunits. A solution destabilization approach (e.g. with chaotropic solvents), and/or a simple crosslinking experiment e.g. with formaldehyde or glutaraldehyde, might show the complete absence of homo-dimers and therefore reinforce the proposed model. Have the authors considered or attempted to perform such expts? If yes, and no additional information could be gained, then I would agree that the presented data is sufficient, but otherwise a discussion of this approach and its results should be included.

Subject to addressing this point, as well as a few more minor points (below), I recommend publication of this excellent work in Nat Comms.

Additional comments:

"a more accurate molecular mass determination, 211 kDa"

When discussing MS data, this suggests very poor accuracy, so I would argue with more precise values

While ion mobility data is shown and CCS values determined, they don't seem to play much of a role for the further discussion here. What additional information can you extract from this parameter?

While the figures highlight the different peak series by utilizing the IM dimension in the 2D plots, it is sometimes also helpful (e.g. in fig. 3) to be able to discern small mass differences and shoulders in peaks, e.g. to estimate metal binding, and I would ask the authors to consider if IM separation

is always necessary to show, and if additional panels can be included to show m/z detail (or exchanged with suppl. figs.).

"two-dimensional spectra with combined resolving power from two techniques" - that is a very odd way of saying it, each dimension has its own resolving power but the combination benefits from an additional, orthogonal dimension (which should be exploited more here - or is it used as more of a filter only)?

I assume that no staining was necessary in EM, but how "true" are the diameters for the particles and how valuable is the comparison with CCS-derived diameters?

It should be stated explicitly where CID and SID were performed - in the trap or transfer stage?

"often causes minimal unfolding, so these metals are more likely to remain bound" - yes, but is there evidence that the full complement of metal remains, or is it just less likely to dissociate? See also different behaviour for E/F. Are any free Cu species detected during SID, e.g. as CuOH+?

The figure legends are too long.

Reviewer #3 (Remarks to the Author):

Molecular mechanisms of biological manganese oxidation are important to the field of environmental biogeochemistry. These mechanisms are highly significant because of the unusually reactive, negatively-charged Mn oxide products have an outsized influence on the cycling of many other trace metals. The Mn-oxide products can serve as virtual "sponges" for many different metallic cations in the environment. Thus, the focus in this work on the molecular structure of a unique manganese-oxidizing enzyme complex (MnxG) has relevance far beyond the biology of the organism which carries it. This work, along with several recent related papers from the Tebo laboratory focusing on studies of the MnxG complex, are major contributions that greatly add to our understanding of biological manganese oxidation process.

The presentation of results and discussion is succinct and to the point. However, given the obvious complexity of the MnxG system, I would have liked to have been more informed about the relevance of this Mn-oxide generating system to the metabolism or survival of the *Bacillus*. What is the evolutionary advantage of Mn oxide formation? Why is it associated in this organism with sporulation? These questions may not have definitive answers, but a bit of speculation would be in order. Perhaps, they have been addressed elsewhere, but a brief summary statement here would be most satisfying to a biologically-oriented reader.

Response to reviewers (original comments in gray color, response in black):

Reviewer #1 (Remarks to the Author):

The manuscript by Romano et al. describes a study that examines the protein structure, stoichiometry and resulting Mn oxides from a complex Mn-oxidizing protein complex (Mnx). The manuscript was very well written and the flow was logical. The data presented was well justified and the authors' conclusions are within the scope of their data. Bacterial Mn-oxidizing proteins have been extremely difficult to study. To date, the Mnx complex is the only via option to get a high quality, purified protein. Thus, this study is very unique as it presents the best structural data on a Mn-oxidizing protein (even though many bacteria are able to oxidize Mn). In addition, the data helps provide important insight into how a spore-forming bacteria is able to generate Mn oxide particles via a protein without that protein getting engulfed with solid phase particles.

We thank the reviewer for positive comments.

Reviewer #2 (Remarks to the Author):

The manuscript by Romano et al. addresses a complex and demanding biological system, Mnx enzyme assemblies which synthesize Manganese Oxide nanoparticles. "Standard" structural or molecular approaches are not expect to give straightforward insights into the structure and function of these complexes, and the choice of cutting-edge MS approaches combined with EM is appropriate. I think this work demonstrates also how polydisperse and otherwise untractable complexes can be investigated, and sets a good example for how novel MS approaches (in this case SID) can be utilized to unravel complex composition and architecture. I am convinced of the findings and agree with the data interpretation, with the (ample) evidence provided, but feel that maybe the inclusion of some minor, additional evidence from other, more readily available MS data would round the story nicely and confirm the otherwise somewhat vague and tentative conclusions, particularly regarding the alternating arrangement of MnxE/F subunits. A solution destabilization approach (e.g. with chaotropic solvents), and/or a simple crosslinking experiment e.g. with formaldehyde or glutaraldehyde, might show the complete absence of homo-dimers and therefore reinforce the proposed model. Have the authors considered or attempted to perform such expts? If yes, and no additional information could be gained, then I would agree that the presented data is sufficient, but otherwise a discussion of this approach and its results should be included.

We agree with the reviewer that some readily available MS methods could help strengthen the conclusions in the manuscript and have in fact performed several experiments with solution destabilization and crosslinking. Unfortunately, these experiments did not yield meaningful results regarding the quaternary structure of Mnx.

Mnx is a very stable protein and requires unusually harsh conditions to be denatured. These characteristics were helpful in purifying Mnx from *E. coli* (where heating to 70° C was used to decompose host proteins), but it can be difficult to selectively denature the Mnx complex. We also

attempted denaturation with high concentrations of urea, but no discernable unfolding was observed. For solution destabilization with MS detection, Mnx is relatively stable under mild denaturation conditions and tends to precipitate in a high percentage of organic solvent. After introduction of denaturant to Mnx solution, we were able to detect only impurity proteins, MnxE monomers, and truncated MnxE monomers.

We also attempted to disrupt Mnx with acetonitrile and 500 mM ammonium acetate (high ionic strength). Similarly to using methanol, we could resolve species related to MnxE monomer with acetonitrile, but Mnx tended to precipitate in a high percentage of acetonitrile. For high ionic strength treatment, we still see mostly intact Mnx complex, with some minor species that appeared to have originated from low abundance impurities and could not be matched to Mnx proteins. Hence, we were not able to extract useful information regarding the quaternary structure of Mnx from these experiments. The emerging species under destabilizing conditions appear to originate mostly from partially modified Mnx (*i.e.* truncated MnxE) and/or impurities. It is likely that the native form of Mnx is more stable than the other components in the sample, making it difficult to capture the substructure of Mnx using typical solution destabilization approaches.

The last few sentences in the first subsection of the results have been revised to capture these results (added text in red):

“Attempts to partially destabilize Mnx complex in solution and detect released sub-complexes were unsuccessful (data not shown). We were, however, able to separate the putative MnxEF assembly *in-situ* from the 211 kDa Mnx complex by gas-phase activation and determine its quaternary structure by IM-MS.”

We also attempted chemical crosslinking experiments using BS2G and BS3 crosslinkers (targeting amine groups), but were unable to identify any inter-crosslinked peptides between any of the 3 different subunits (MnxE, MnxF, and MnxG) with high confidence after in-solution digestion presumably due to low crosslinking efficiency. We suspect there might be limited primary amine groups in proximity for crosslinkers, and it may require additional optimization of conditions, or a different crosslinker altogether, to obtain useful information. Following the reviewer’s suggestion, we performed crosslinking experiments with formaldehyde and glutaraldehyde. The denaturing SDS-PAGE indicated low efficiency of crosslinking that would result in capturing the MnxE/MnxF multimers (*i.e.* very faint or no bands corresponding to the mass range of interest. Increasing formaldehyde and glutaraldehyde concentrations induced excessive crosslinking and even caused protein precipitation in the case of glutaraldehyde.

We added a comment in the manuscript to describe the difficulties encountered in crosslinking experiments (added text in red):

“Mapping the interactions between MnxE and MnxF using chemical crosslinking proved to be challenging, presumably due to low crosslinking efficiency. Still, the subunit connectivity determined from SID data, combined with protein modeling and docking simulations, led to a tentative structural model of the Mnx complex, shown in Figure 4b (more details in Supplementary Fig. 5).”

We also expanded the Supplementary Information (end of Supplementary Methods 2) to include the results using formaldehyde and glutaraldehyde while discussing the docking simulation (added text in red):

“To generate models at higher confidence in the absence of high resolution data from conventional techniques, it is essential to obtain more constraints from experiments. We will pursue this information through additional labeling experiments, crosslinking, and computational efforts exploring larger conformational space. **Our initial crosslinking experiments with bis(sulfosuccinimidyl) 2,2,4,4-glutarate (BS2G) and bis(sulfosuccinimidyl)suberate (BS3) did not generate confident identifications for inter-subunit crosslinks due to low efficiency of crosslinking. Crosslinking experiments with paraformaldehyde and glutaraldehyde were also not able to effectively capture multimers of MnxE and MnxF, and using high concentrations of these reagents resulted in excessive crosslinking and caused precipitation.**”

Subject to addressing this point, as well as a few more minor points (below), I recommend publication of this excellent work in Nat Comms.

Additional comments:

"a more accurate molecular mass determination, 211 kDa"

When discussing MS data, this suggests very poor accuracy, so I would argue with more precise values

We understand the reviewer's concern that the molecular mass determination should be accurate enough to include more significant figures. However, as mentioned in the Figure 1 caption and Supplementary Fig. 2 caption, the mass uncertainty primarily came from the protein complex heterogeneity. The broad peak yields uncertainties of mass of a few hundred Da justifying the lack of significant figures when reporting the measured molecular mass. In response to the reviewer's comment, we revised the sentence to read (beginning of Results section):

“Relative to the ~ 230 kDa previously estimated by SDS-PAGE and SEC, native mass spectrometry of the Mnx complex (electrosprayed in 100 mM ammonium acetate, pH 7, to preserve noncovalent assembly) allowed a more accurate molecular mass determination of 211216 Da \pm 0.8 kDa. The uncertainty is due to the large compositional heterogeneity of the protein complex.”

While ion mobility data is shown and CCS values determined, they don't seem to play much of a role for the further discussion here. What additional information can you extract from this parameter?

We used the CCS values of Mnx complex (unactivated) and released MnxE/MnxF monomers to help restrict the modeling as discussed in Supplementary Fig. 5 caption. We originally attempted correlating experimental CCS values with TEM diameter. However, we initially did not include this in the discussion

because we felt the results were highly speculative. We provide a more comprehensive response below in response to another comment related to comparing EM data with CCS data.

While the figures highlight the different peak series by utilizing the IM dimension in the 2D plots, it is sometimes also helpful (e.g. in fig. 3) to be able to discern small mass differences and shoulders in peaks, e.g. to estimate metal binding, and I would ask the authors to consider if IM separation is always necessary to show, and if additional panels can be included to show m/z detail (or exchanged with suppl. figs.).

We decided to show the 2D plots for Figure 3 as the main figure for simplicity. The figure was designed to quickly show the many types of subcomplexes released in SID for quaternary structure determination. Within a simple 2D plot (Figure 3d), nearly all species can be resolved directly and be compared with other conditions (MS, CID, SID) side-by-side. For SID spectra, multiple species can partially overlap and the relative abundances of each species vary broadly, making visualization difficult in a single 1D mass spectrum (only able to see the most abundant species: M_nxE/M_nxF monomers, dimers, M_nxG , and remaining undissociated M_nx).

We included Figure 5 to clearly show metal binding on M_nxE/M_nxF subunits along with the discussion. As the reviewer may have also noticed, we did include the extracted mass spectra as a multiple panel figure in Supplementary Information (Supplementary Fig. 4) as a more detailed demonstration for characterizing metal binding. To better display the metal binding species, we expanded the m/z axis for Supplementary Fig. 4 and annotated the major Cu binding species (copied below).

Revised Supplementary Fig. 4 to better display the metal binding species addressing the reviewer's comment. Caption updated accordingly in the revised supplementary information. In short, the blue circles show the number of bound Cu assigned based on the mass shift from apo species. It is noted that for the EDTA treated sample, the peaks carrying extra mass originated from post-translational modifications (based on separate high resolution data as discussed below).

For species larger than MnxEF trimer, the broadness of the peaks (due to sample heterogeneity) did not allow in-depth characterization of metal binding. Even in our follow-up study using FTICR MS, we were only able to resolve species with mass values up to MnxEF hexamer (data shown below, figure for manuscript under review for Journal of the American Society for Mass Spectrometry). For these reasons herein we focused on metal binding on smaller species, *i.e.*, the released MnxE and MnxF monomers. We believe the revised Supplementary Fig. 4 provides sufficient details for readers who are interested in seeing the mass spectra.

[REDACTED]

"two-dimensional spectra with combined resolving power from two techniques" - that is a very odd way of saying it, each dimension has its own resolving power but the combination benefits from an additional, orthogonal dimension (which should be exploited more here - or is it used as more of a filter only)?

The term "with combined resolving power from two techniques" has been removed.

I assume that no staining was necessary in EM, but how "true" are the diameters for the particles and how valuable is the comparison with CCS-derived diameters?

In fact, due to the very low contrast of these particles, we used a TEM negative stain NanoW ©. This stain, unlike the traditional uranyl acetate, is buffered for osmotic balance with the stained material in circa-neutral conditions, and it is generally considered inert to the stained material (i.e. not causing material contraction). The smaller particle size relative to the calculated model can be attributed mostly to the particles' conformational heterogeneity, as clearly illustrated in the HR TEM images. The particles are not perfectly spherical, and in addition, they have irregular surface features, so measuring the particles was a challenge. We used a TEM analytical software package (TIA by FEI Co.) based on our best judgement for measuring particle diameter by comparing different methods (directly measuring diameters versus drawing circles and ovals to encompass a particle and measuring the diameters of those). A computational single particle reconstruction scheme would be the best solution for the 3D model, but it was not applicable on the Mnx complex due to its small size.

We attempted to compare EM diameters with CCS-derived diameters. Assuming the protein is a perfect sphere, the CCS is identical to the projection area of the sphere (plus collision gas radius) and can be converted to diameter of the particle (Hall et al. *Structure* **2012**, *20*, 1596-1609). We suspect that the irregular surface features of the Mnx complex are effectively probed by ion mobility measurements but not by the particle size measurement in TEM, which is based on projection area. The diameter of the Mnx complex estimated from CCS of Mnx (10.1 nm) is significantly larger than the average diameter estimated from TEM (7.91 nm). The MnxEF particle, however, showed a consistent diameter of about 6.8 nm for both methods. We suspect there might be some preferred orientation for Mnx on the surface of the TEM grids so only the dimension along one axis was captured by TEM, causing the discrepancy. We decided to not include this discussion because it is highly speculative at this point and may cause confusion.

It should be stated explicitly where CID and SID were performed - in the trap or transfer stage?

CID and SID were both performed in the trap ion guide for the data presented. Experimental details have been added in the method section for clarification (added words in red):

“Briefly, the modified mass spectrometer allows the electrosprayed native protein complexes to be activated via either surface collision (surface induced dissociation, SID) or neutral gas collision (collision induced dissociation, CID) for dissociation before ion mobility separation **Activation in the trap ion guide of the instrument occurs after quadrupole m/z selection and before IM. The final step involves mass detection by the time-of-flight mass analyzer.**”

"often causes minimal unfolding, so these metals are more likely to remain bound" - yes, but is there evidence that the full complement of metal remains, or is it just less likely to dissociate?

We don't want to claim that the full complement of metal remains because we do expect that excess internal energy may cause loss of very weakly bound metal/ligand without unfolding of the protein (*i.e.*, weak binding to the surface of the protein). As mentioned in Supplementary Discussion 2, 10-15 Cu per Mnx complex were determined based on ICP OES (Butterfield et al. *Metallomics* 2017, **9**: 183- 191). At least 4 of these Cu atoms are suspected to bind within the MnxG subunit, as is customary for MCO proteins. Several putative Cu binding sites were determined previously (Dick et al. *Appl Environ Microbiol* 2008, **74**: 1527-1534). Given that we can observe almost ~10 Cu in MnxEF hexamer and that MCOs (like MnxG) contain at least 4 Cu, (Solomon et. al. *Chem Rev* 1996, **96**: 2563-2606) it appears that the majority of bound Cu is maintained.

We have expanded the Supplementary Discussion 2 to show that at least qualitatively we maintained the majority of Cu in the native MS results:

“Nonetheless, for Mnx prepared in HEPES, we estimate ~10 Cu bound to MnxEF hexamer based on SID results and it is well-established that MCOs (like MnxG) contain at least 4 Cu¹⁴. The stoichiometry is qualitatively consistent with our previous ICP result. Yet further improvements in high resolution native MS are required to better define the Cu binding stoichiometry in MnxG.”

See also different behaviour for E/F.

In regard to the different behavior of MnxE/MnxF, we included a brief discussion in Supplementary Discussion 2. We believe the major Cu binding site in MnxE is localized at the N-terminus, which may not require a tertiary fold, and thus can be maintained even when the protein is largely unfolded. This conclusion is based on the observation that a truncated MnxE (*i.e.*, missing the first 9 residues) has largely diminished Cu binding as discussed in our recently submitted manuscript describing high resolution FTICR spectra (submitted as related manuscript with our initial submission, this study was presented at a conference and the proceeding was cited as reference 54), where we were able to unambiguously assign most of the species detected. Briefly, metal binding behavior is directly correlated to the protein structure, and these differences can be nicely explained using native MS experiments in combination with SID.

A sentence is added (in red color) in the discussion to mention our reasoning behind the different behaviors of MnxE/MnxF:

“Overall, MS data suggest MnxE binds copper with a high affinity and a primarily 1:1 stoichiometry. MnxF can bind multiple copper atoms but exhibits lower affinity to copper than MnxE. **Differences in metal binding behavior between MnxE and MnxF may be due to the location of the Cu binding sites within the protein** (Supplementary Discussion 2).”

The related discussion in Supplementary Discussion 2 is also updated to include more details based on reference 54:

“We hypothesize that this variability results from the location of these metals' binding sites within the secondary and/or tertiary structure of their subunits. The major metal binding site in MnxE is likely

localized at several residues at the N-terminus which may not require a specific tertiary fold, therefore the metal binding is not directly affected by unfolding”

Are any free Cu species detected during SID, e.g. as CuOH^+ ?

The instrument does not allow us to detect low mass ions such as free Cu^{2+} (the minimum low mass range is at $m/z = 50$). The reviewer raised an interesting point that we may be able to detect CuOH^+ . We expanded the low mass of the acquisition range to m/z 50 for both CID and SID, but we were not able to detect Cu related species released during SID or CID. We suspect Cu is likely to be released (if any) as Cu^{2+} , which is outside the detection range of the instrument. It would be an interesting direction to simultaneously detect small metal/ligand and protein complexes for future work.

The figure legends are too long.

We tried to provide extensive details in figure legends for maximum clarity. We have revised the caption text following the requirements of Nature Communications. All figure legends are now 350 words or less with a short title. The supplementary figure legends have also been revised.

Reviewer #3 (Remarks to the Author):

Molecular mechanisms of biological manganese oxidation are important to the field of environmental biogeochemistry. These mechanisms are highly significant because of the unusually reactive, negatively-charged Mn oxides products have an outsized influence on the cycling of many other trace metals. The Mn-oxide products can serve as virtual "sponges" for many different metallic cations in the environment. Thus, the focus in this work on the molecular structure of a unique manganese-oxidizing enzyme complex (MnxG) has relevance far beyond the biology of the organism which carries it. This work, along with several recent related papers from the Tebo laboratory focusing on studies of the MnxG complex, are major contributions that greatly add to our understanding of biological manganese oxidation process. The presentation of results and discussion is succinct and to the point. However, given the obvious complexity of the MnxG system, I would have liked to have been more informed about the relevance of this Mn-oxide generating system to the metabolism or survival of the *Bacillus*. What is the evolutionary advantage of Mn oxide formation? Why is it associated in this organism with sporulation? These questions may not have definitive answers, but a bit of speculation would be in order. Perhaps, they have been addressed elsewhere, but a brief summary statement here would be most satisfying to a biologically-oriented reader.

We appreciate the reviewer's comment about the significance of our work. In response to the reviewer's concern, we expanded the introduction to include more details along with some speculations on the biological relevance of these Mn-oxide generating systems:

“While research has established the importance of microbial Mn oxidation on geochemical Mn cycling, questions remain about the physiological benefits organisms derive from this process. Some bacteria may employ Mn oxidation to make them more resilient to oxidative stress³. This hypothesis was inspired by work demonstrating that increased intracellular Mn levels make *Deinococcus radiodurans* more resilient to radiation⁴, and also by numerous studies demonstrating the ability of Mn containing proteins and small molecules to reduce intracellular superoxide levels^{5, 6, 7, 8, 9, 10}. Other benefits may include disposing of excess O₂ by transforming it into part of an insoluble oxide mineral, forming a Mn oxide crust around a microbe which could deter predation or viral attack, or as a mechanism for degrading and gaining energy from natural organic matter². This latter hypothesis is appealing since some of the enzymes that oxidize Mn(II) are related to laccases, enzymes that are involved in lignin decomposition^{11, 12}. Additionally, Mn oxides themselves oxidatively degrade natural humic substances producing low molecular weight organic compounds that are potential substrates for microbial growth².

Recently, several enzymes involved in bacterial Mn(II) oxidation have been described. The Mn oxidases that have been characterized thus far belong to two families of proteins: the animal heme peroxidases (AHPs) and the multicopper oxidases (MCOs). Generally, these enzymes appear to catalyze the oxidation of Mn(II) to Mn(IV) via a transient Mn(III) intermediate^{13, 14, 15}. The Mn(II)-oxidizing AHPs have been described in several alphaproteobacteria^{13, 16, 17} and in the gammaproteobacterium *P. putida* GB-1¹⁸. Some AHPs appear to oxidize Mn(II) directly^{13, 17}, while that of *Roseobacter* sp. AzwK-3b is thought to produce O₂⁻, which oxidizes Mn(II)¹⁶.

MCOs are of interest to biochemists and environmental scientists because of their roles in various processes, including degradation of complex carbon sources, metal homeostasis and metabolism, and a variety of electron transfer reactions^{11, 12, 19, 20, 21, 22, 23, 24}. MCOs that catalyze Mn oxidation have been described in several model organisms, including species of *Bacillus*^{25, 26}, *Pseudomonas*²⁷, *Leptothrix*²⁸ and *Pedomicrobium*²⁹.”

REVIEWERS' COMMENTS:

Reviewer #2 (Remarks to the Author):

The authors have addressed all my questions and concerns well, and I am satisfied with their responses. I recommend this paper now for publication.